

# Density and diversity of macroinvertebrates in Colombian Andean streams impacted by mining, agriculture and cattle production

Ana M. Meza-Salazar[1,2], Giovany Guevara[3], Lucimar Gomes-Dias[1] and Carlos A. Cultid-Medina[4,5]

[1] Facultad de Ciencias Exactas y Naturales, Grupo de Investigación BIONAT, Universidad de Caldas, Manizales, Caldas, Colombia
[2] Department of Applied Ecology, North Carolina State University, College of Agriculture and Life Sciences, North Carolina, NC, United States of America
[3] Facultad de Ciencias, Departamento de Biología, Grupo de Investigación en Zoología (GIZ), Universidad del Tolima, Ibagué, Tolima, Colombia
[4] Red de Diversidad Biológica del Occidente Mexicano, Instituto de Ecología, INECOL A.C., Centro Regional del Bajío,, Pátzcuaro, Michoacán, México
[5] CONACYT, Ciudad de México, Ciudad de México, México

Corresponding author
Carlos A. Cultid-Medina,
carlos.cultid@inecol.mx

## ABSTRACT

**Background**. Mining, agriculture and cattle production are activities that threaten the quality and quantity of water resources in the Colombian Andes. However, many drainage basins in this region have not been subjected to simultaneous evaluation of the impact these activities have on the density, diversity and composition of aquatic macroinvertebrates (AMI). The first two of these ecological variables are expected to decrease drastically from zones with no apparent impact towards areas with anthropogenic activity, which areas with mining will present the most impoverished AMI community.

**Methods**. We evaluated the density, diversity and composition dissimilarity of AMI in streams impacted by gold mining, agriculture and cattle production. Two reference streams were also studied. Six benthic samplings were conducted bimonthly (Feb 2014–Feb 2015) using a Surber net. Water samples were taken in order to make environmental evaluation among the aforementioned streams, including hydrological, physicochemical and bacteriological parameters (HPCB). Diversity was evaluated as the effective number of RTUs—recognizable taxonomic units—by comparing the richness, typical diversity, and effective number of the most abundant RTUs. Compositional dissimilarity was examined with nMDS and CCA analysis.

**Results**. A total of 7,483 organisms were collected: 14 orders, 42 families and 71 RTUs. Our prediction regarding the density and diversity of AMI (Reference > Cattle production > Agriculture > Mining) was partially fulfilled, since the agriculture-dominated stream presented a more impoverished AMI community than that of the gold mining stream. However, these streams presented lower diversity than the cattle production and reference streams, and the AMI density only differed significantly between one reference stream and the agriculture stream. The AMI composition in the agriculture-dominated stream clearly differed from that of the other streams.

**Discussion**. The observation of a more impoverished AMI community in agricultural production areas compared to those with mining or cattle production may reflect the importance of the remaining riparian vegetation, which was scarce at the stream with agricultural activity. Moreover, the low diversity, and mainly the reduced AMI richness, in the agriculture stream coincided with the absence of insect genera are intolerant to deterioration of the biological and physicochemical conditions of the water (e.g. *Anacroneuria*).

**Conclusions**. The results suggest that the local impact of agricultural activities may be of equal or greater magnitude than that of mining in terms of AMI density, diversity and composition, in the Colombian Andean riverscape. Future studies should systematically evaluate, throughout the annual cycle, the relative effects of the productive land use, the remaining native vegetation cover and the consequent changes in the HPCB parameters of the water on AMI communities in Colombian Andean basins.

# INTRODUCTION

Over the last four decades, pressure on lotic systems has increased in an accelerated manner at global level as a consequence of the rapid expansion of areas of anthropogenic exploitation (*Haddeland et al., 2014*). The main threats to global freshwater diversity include overexploitation, water pollution, flow modification, habitat destruction/degradation and invasion by exotic species (*Dudgeon et al., 2006*; *Vörösmarty et al., 2010*; *Malaj et al., 2014*; *Reid et al., 2019*). Continuous overuse increases the deforestation rate of riparian vegetation and thus increases runoff, causing changes in the stream morphology and consequently the habitat degradation. These changes affect the physicochemical parameters of the water, contributing to the impoverishment of aquatic biodiversity (*Etter & Wyngaarden, 2000*; *Zapata et al., 2007*; *Larson, Dodds & Veach, 2019*).

In particular, different studies have shown how mining, agricultural and cattle production threaten the quality of, and access to, hydric resources (*Lobo et al., 2017*; *Grudzinski & Daniels, 2018*; *Mwangi et al., 2018*). In Colombia, agriculture, cattle production and mining have put both the quality and availability of hydric resources at risk over the last decade (*Chará-Serna et al., 2015*; *Villada-Bedoya et al., 2017*; *Villada-Bedoya, Triana-Moreno & G-Dias, 2017*; *Ramírez et al., 2018*). These activities threaten the lotic systems of the Andes, where the human population of the country is concentrated (*Murtinho et al., 2013*; *Guevara, 2014*; *Chará-Serna et al., 2015*).

In recent decades, aquatic macroinvertebrates (AMI) have been widely studied as effective bioindicators in the evaluation of the impact of human activities on freshwater ecosystems (e.g., *González, Basaguren & Pozo, 2003*; *Prat et al., 2009*; *Buss et al., 2015*). At both community and population level, these organisms are highly sensitive to changes in the physicochemical properties of the water and to habitat quality (*Roldán, 2003*; *Alonso & Camargo, 2005*; *Roldán-Pérez, 2016*; *Carter, Resh & Hannaford, 2017*). Different studies in the Neotropics have evaluated the effects of mining, agricultural and cattle
production activities on AMI (e.g., *Villamarín-Flores, 2008*; *Hepp et al., 2010*; *Mesa, 2010*; *Miserendino & Masi, 2010*; *Ordóñez, 2011*; *Egler et al., 2012*; *Terneus, Hernández & Racines, 2012*; *Fierro et al., 2015*). In recent years, studies exploring the effects of cattle production, agriculture and mining activities on the AMI communities have increased in Colombia (e.g., *Chará & Murgueitio, 2005*; *Feijoo, Zuñiga & Camargo, 2005*; *Galindo-Leva et al., 2012*; *Gómez, 2013*; *Villada-Bedoya, Triana-Moreno & G-Dias, 2017*; *Ramírez et al., 2018*), and have documented changes in the ecological attributes of the AMI as a consequence of anthropogenic alterations to inland water resources.

In the case of species richness, greater values have been recorded in reference streams compared to those with an influence of mining, agriculture or cattle production (*Feijoo, Quintero & Fragoso, 2006*; *Egler et al., 2012*; *Terneus, Hernández & Racines, 2012*), mainly due to the reduction in riparian vegetation and introduction of polluting substances. In terms of abundance (or density), some studies have recorded greater values in sites with anthropogenic impacts compared to those with greater quantities of surrounding vegetation (*Chará & Murgueitio, 2005*; *Miserendino & Masi, 2010*). This is due to the dominance of certain taxa, as has been observed in streams dominated by agriculture (*Egler et al., 2012*) and cattle production (*Mesa, 2010*; *Giraldo et al., 2014*). Likewise, AMI composition also presents important differences between streams with and without evident anthropogenic impact (*Hepp et al., 2010*).

Among the activities that most degrade the aquatic ecosystem, mining has been considered to have serious effects on water quality and quantity due to mining wastes and the ecological impairment of habitats (*Cidu, Biddau & Fanfani, 2009*; *Wright & Ryan, 2016*). Channel diversion and the removal of organic matter and sediments affect the availability of refuge and food for benthic organisms, making it difficult to colonize and/or recover long-term communities (*Milner & Piorkowski, 2004*). However, few studies have conducted simultaneous evaluation of the effects of mining, agriculture and cattle production in Andean streams (*Villada-Bedoya et al., 2017*; *Villada-Bedoya, Triana-Moreno & G-Dias, 2017*; *Ramírez et al., 2018*). It is important to recognize that the Neotropical region presents a wide variety of climatic conditions and habitat heterogeneity, for which reason the diversity patterns are dynamic and can be influenced by many factors (land use, local geography, availability of riparian vegetation, among others). Further knowledge of the patterns of AMI density and diversity is therefore necessary (*Guevara, 2014*; *Buss et al., 2015*).

This study evaluated the density, diversity and compositional dissimilarity of the AMI in contrasting headwater streams of the Colombian Andes; two near-pristine streams, and one stream in zones with agricultural, cattle production and gold mining activities, in the Chinchiná river basin (Caldas, Colombia). According to the assumed impact of each productive land use, we expected that: (1) AMI density will increase from the two reference streams to those of agriculture, cattle production and mining, (2) this increase in density will reflect an increased dominance of taxa that are tolerant to the water pollution, and (3) a maximum impoverishment of AMI diversity will be found in the zone with gold mining activity.

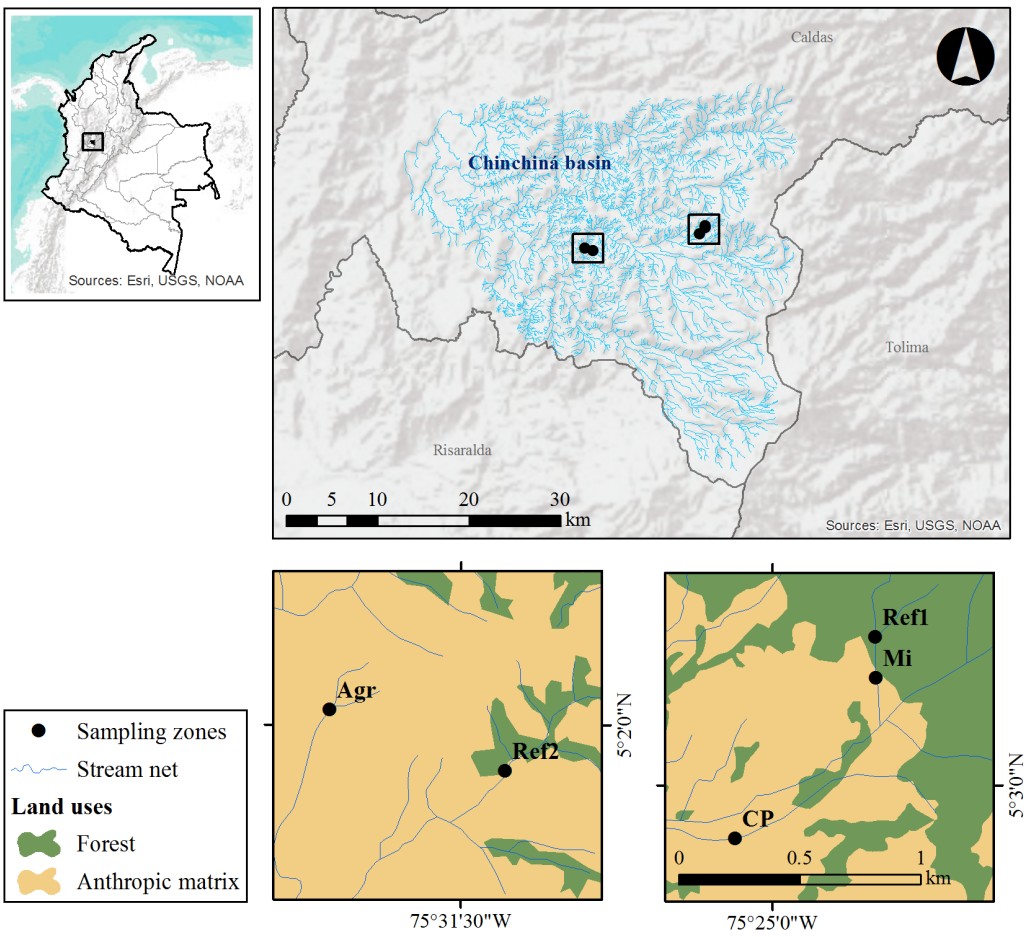

**Figure 1** Study area and sampling zones located on the western slope of the central cordillera of the Colombian Andes, in the Chinchiná river basin (Caldas, Colombia).

# MATERIAL AND METHODS

## Study area

The selected streams are located on the western slope of the central cordillera of the Colombian Andes, in the municipalities of Villamaría and Manizales (Caldas, Colombia), and are tributaries of the Chinchiná river basin. Five sampling zones were selected, three of these zones had productive impacts (agriculture, cattle production and gold mining) while the other two were of reference condition; i.e., streams with no evident local anthropogenic impacts (Fig. 1). In each zone, AMI sampling was carried out along 100 m of the streams.

Reference 1 (Ref1): Located in the stream La Elvira, sector Maltería (Manizales: 05°03′10.9″N, 75°24′33.6″W) at 2766 m asl. This area presents riparian vegetation of greater than 15 m in width, mainly comprising herbaceous plants, shrubs and trees. The most representative plant species include *Aiouea* sp., *Clethra revoluta* Ruiz and Pav., *Dunalia solanacea* Kunth, *Miconia superposita* Wurdack and *Verbesina nudipes* S.F. Blake.

Reference 2 (Ref2): Located in the stream La Floresta (Villamaría: 05°1′42.1″N, 75°31′10.9″W) at 1,720 m asl, close to agricultural zones and used as an area of recreation. Its riparian vegetation is more than 15 m in width and presents elements characteristic of conserved forest (*Guariguata & Ostertag, 2002*), such as large trees of the families Moraceae (*Ficus* sp., *Coussapoa duquei* Standley), Lauraceae (*Nectandra* sp.) and Boraginaceae (*Cordia panamensis* L. Riley).

Cattle production (CP): Located in the stream Cimitarra, sector Maltería (Manizales: 05° 04′32.0″N, 75° 24′0.60″W) at 2,550 m asl. It is surrounded by grazing pastures, although the cattle have no access to the stream due to the presence of a strip of vegetation of approximately 3 m in width on both banks, which is dominated by species of early succession such as: *Baccharis latifolia* Ruiz and Pavón, *Miconia superposita* Wurdack, *Rubus glaucus* Benth, *Aphelandra acanthus* Nees, *Solanum phaeophyllum* Werderm and *Tibouchina lepidota* Bonpl. In addition, two introduced plant species were recorded: *Pennisetum clandestinum* Hochst. ex Chiov (Poaceae), cultivated as pasture, and *Lachemilla orbiculata* Ruiz & Pav. (Rosaceae), a plant species abundant in grazing pastures of cold climates (*Vargas, 2002*).

Agriculture (Agr): Corresponding to the stream "Don Alonso" (Villamaría: 05°01′50.79″N, 75°31′39.59″W) at 1849 m asl. The riparian vegetation is practically absent (only small shrubs, grasses, and sparse herbaceous plants persist). In addition, this area also has closer vegetable gardens in which the following species are cultivated in alternation with the following species: *Brassica oleracea* var. *capitata* Linnaeus and *Brassica oleracea* var. *italica* Linnaeus, *Sechium edule*. (Jacq.) Sw., *Musa velutina* H. Wendl. and Drude, *Guadua angustifolia* Kunth, *Urera baccifera* (L.) Gaudich., *Piper* cf. *crassinervium* Kunth, *Montanoa quadrangularis* Schultz Bipontianus, *Cecropia angustifolia* Trécul.

Mining (Mi): Located on the stream La Elvira (Manizales: 05°03′4.4″N, 75°24′33.1″W) at 2725 m asl. Its riparian zone is fragmented by land use change through activities of auriferous mining extraction using mercury. The stream presents vegetation comprising grazing pastures and secondary forest with an approximate width of 1 to 2 m, dominated by grasses (*Pennisetum clandestinum* Hochst. ex Chiov), herbaceous plants (*Coniza bonariensis* (L.) Cronquist), *Hypochaeris radicata* L., *Taraxacum officinale* G. H. Weber ex Wigg, *Lachemilla orbiculata* Ruiz and Pavón, *Plantago major* L., (*Gunnera brephogea* Linden & André) and some juvenile trees (*Baccharis latifolia* Ruiz and Pavón and *Miconia* cf *theaezans* Bonpl.).

## Collection of organisms

The AMI density (ind/m$^2$) and diversity, and composition of RTUs, were evaluated based on Rapid Bioassessment Protocols (RBP) (*Barbour et al., 1999*). We used a Surber net (30 × 30 cm, mesh size 250 μm) with three replicates in each of three substrates (leaf litter, rock and sediment; *Aazami et al., 2015*) during six sampling events per stream (between February 2014 and February 2015), giving a total of 54 samples per zone. The collected material was fixed in vials containing 96% alcohol and the AMI identified to the lowest practical taxonomic level (usually genus) using the taxonomic keys of *Merritt & Cummins (1996)*, *Domínguez et al. (2006)*, *Gutiérrez & Dias, 2015* and *Domínguez &*

*Fernández (2009)*. Specimen collection permits were regulated by Resolution 1166 of October 9th, 2014, issued by the National Environmental Licenses Authority (ANLA, by its Spanish acronym) of Colombia and by Decree 1376 of June 27th, 2013 from the Colombian Ministry of Environment and Sustainable Development. The material was deposited in the Entomological Collection of the Programa de Biología of the Universidad de Caldas—CEBUC (certified collection under register: No 188 in the Registro Nacional de Colecciones Biológicas—RNC administered by Instituto de Investigación de Recursos Naturales Alexander von Humboldt).

## Hydrological, physicochemical and bacteriological parameters

The environmental characterization of the sampling streams involved 27 different hydrological, physicochemical and bacteriological (HPCB) parameters and elevation (m asl). Among the hydrological parameters, water flow volume ($m^3$/s) was measured in each sampling event and mean precipitation (mm/week) in each month of sampling was recorded (*IDEAM, 2015*). In February, July and November 2014, the following water and stream parameters were measured (*in situ*, Table S1): velocity (m/s), width (m), depth (cm), temperature (Temp, ° C), pH, conductivity (Con, µS/m) and dissolved oxygen (DO, mg/L). Temperature, pH and conductivity were measured with an OAKLON PH/CON 300 multiparameter device, while dissolved oxygen was measured with a Lutron do-5510 dissolved oxygen meter. Water samples were taken and transported to the IQ&A (Ingenieros químicos y asociados S.A., Manizales, Colombia) certified laboratory for determination of the following parameters (Table S1): chlorides (Ch, mg/L), sulphates ($SO_4$, mg/L), nitrites ($NO_2$, mg/L), phosphates ($PO_4$, mg/L), fats and oils (FO, mg/L), biochemical oxygen demand (BOD, mg/L), chemical oxygen demand (COD, mg/L), total dissolved solids (TS, mg/L), total suspended solids (TSS, mg/L), ammoniacal nitrogen ($NH_3$-N, mg/L), aluminum (Al, mg/L), mercury (Hg, mg/L), total iron (Fe, mg/L), lead (Pb, mg/L), cyanide (Cy, mg/L), boron (B, mg/L), *Escherichia coli* (Ecoli, CFU/100 mL) and total coliforms (Tc, CFU/100 mL) (*Chará, 2003*; *Sánchez, 2004*).

## Data analysis

The AMI density values among sampling zones were analyzed with a non-parametric repeated measures Friedman test ($n = 6$ sampling events) and particular differences among streams were identified with a *post-hoc* Nemenyi test (*Zar, 2010*). Diversity was estimated as the effective number of RTUs or diversity order q ($^qD$; *Jost, 2006*):

$$^qD = \left( \sum_{i=1}^{S} p_i^q \right)^{1/(1-q)}$$

Where pi is the relative abundance (proportional abundance) of the *i*-th RTU, S is the number of RTUs and the *q*-value is the order of the diversity. When q=0, richness is obtained. When q $\approx$ 1, the effective number of equally common genera is obtained. This is equivalent to the exponential of the Shannon index of entropy and does not present bias as a result of the presence of either rare or abundant RTUs in the sampling. Finally, when q=2, the value of diversity indicates the effective number of the more abundant RTUs in

the sampling and is equivalent to the inverse of the Simpson index of entropy (*Moreno et al., 2011*).

Since the continuous variable of density was used as an abundance measure, estimation of sample coverage (Ĉn, see *Chao & Jost, 2012*) per stream was not required prior to making the diversity comparisons. In each case, we obtained a completeness of 100% (absence of singletons), and the diversity comparison was therefore made directly with the observed values of $^qD$. The CI 95% of each expression of diversity ($^0D$, $^1D$, $^2D$) was used as a statistical criterion, in which absence of overlap between the CI 95% indicated significant differences between the values of diversity (*Cumming, Fidler & Vaux, 2007*; *Chao et al., 2020*). Estimation of $^qD \pm$ CI 95% was conducted with the package iNEXT of R (*Hsieh, Ma & Chao, 2015*).

By expressing diversity as the effective number of RTUs and making comparisons under the same and maximum sample coverage (100%), the replication principle is met and it is possible to calculate the magnitude of the difference in diversity (MD = Sampling Site 2 / Sampling Site 1) among communities (*Jost, 2006*; *Moreno et al., 2011*). It is thus possible to determine how many times one zone is more or less diverse than another. In addition, comparison of $^qD \pm$ CI 95% under the effective numbers of RTUs eliminates estimation bias due to the high density of certain aquatic insect groups, such as the dipterans (e.g., Chironomidae). It would be impossible to avoid this bias using the classic protocol for the use of rarefaction curves, which relies on a comparison based on minimum sample size or minimum abundance. To evaluate the differences in density and the incidence of dominant taxa tolerant to water contamination, rank-density curves were constructed per sampling zone. On the *x*-axis, RTUs were ranked in descending order according to density (*y*-axis in logarithmic scale). These curves not only allow visualization of the distribution of density among the RTUs but also determination of which taxa disappear or appear and the relative positions they occupy in each sampling area, according to their density. This information, together with the MD, may be more useful for the ecological diagnostic of the effects of anthropogenic impact on water conditions (*Feinsinger, 2001*).

The compositional dissimilarity of AMI RTUs was examined with a non-metric multidimensional scaling (nMDS) based on the Bray-Curtis index (*Quinn & Keough, 2002*). An ANOSIM was used to determine whether the compositional dissimilarity was greater among than within zones, and the contribution of the RTUs to the dissimilarity was subsequently established using a SIMPER (*Quinn & Keough, 2002*). The patterns of density, diversity and compositional dissimilarity were discussed with respect to HPCB parameters. First, we used a Spearman correlation test to examine how changes in AMI density were related to flow and precipitation (Table S2). Secondly, since the HPBC was measured in only three sampling moments (i.e., Feb, Jul, Nov 2014), we performed a CCA analysis to evaluate the association patterns among RTUs, sites and HPBC parameters regarding pair-consecutive AMI sampling events: Feb14+Apr14; Jul14+Sept14; Nov14+Feb15. This temporal grouping of data was also used for the compositional dissimilarity analyses (see above). To avoid collinearity among HPBC parameters, we applied the Variance Inflation Factor (VIF) and HPBC parameters with VIF >10 were thus excluded from the CCA

**Table 1  Number of individuals for each recognizable taxonomic unit (RTU) in each sampling zone.**

| Order | Family | Genera | Ref1 | Ref2 | CP | Agr | Mi |
|---|---|---|---|---|---|---|---|
| Amphipoda | | *Hyalella* | 3 | 1 | 138 | 0 | 1 |
| Arhynchobdellida | Hirudinidae | H1 | 1 | 0 | 0 | 0 | 0 |
| Coleoptera | Dryopidae | Dr1 | 0 | 2 | 0 | 0 | 0 |
| | Dytiscidae | Dy1 | 0 | 0 | 0 | 1 | 2 |
| | Elmidae | *Austrolimnius* | 0 | 1 | 0 | 0 | 0 |
| | | *Cylloepus* | 1 | 16 | 2 | 1 | 3 |
| | | *Disersus* | 0 | 0 | 0 | 0 | 1 |
| | | *Heterelmis* | 3 | 13 | 88 | 0 | 2 |
| | | *Macrelmis* | 2 | 10 | 0 | 0 | 1 |
| | | *Neoelmis* | 0 | 0 | 0 | 0 | 1 |
| | | *Pharceonus* | 0 | 0 | 2 | 0 | 0 |
| | Hydrophilidae | *Hydrophilus* | 0 | 0 | 1 | 0 | 0 |
| | Ptilodactylidae | *Anchytarsus* | 116 | 82 | 122 | 4 | 49 |
| | Scirtidae | Sc1 | 106 | 0 | 22 | 0 | 1 |
| Decapoda | Pseudothelphusidae | *Strengeriana* | 0 | 1 | 0 | 4 | 0 |
| Diptera | Blephariceridae | *Limonicola* | 13 | 0 | 2 | 0 | 9 |
| | | *Paltostoma* | 0 | 0 | 5 | 0 | 0 |
| | Ceratopogonidae | *Bezzia* | 7 | 0 | 0 | 0 | 3 |
| | Chironomidae- Subfamily Chironominae | Ch1 | 10 | 268 | 41 | 2 | 10 |
| | | *Polypedilum* | 0 | 0 | 0 | 0 | 3 |
| | | *Riethia* | 1 | 0 | 0 | 0 | 0 |
| | Chironomidae-Subfamily Tanypodinae | Tany1 | 0 | 53 | 0 | 1 | 1 |
| | Chironomidae-Subfamily Orthocladiinae | Oth1 | 75 | 328 | 31 | 5 | 271 |
| | Chironomidae-Subfamily Podonominae | *Podonomus* | 10 | 0 | 2 | 0 | 0 |
| | Dixidae | Dix1 | 0 | 1 | 2 | 0 | 0 |
| | Dolichopodidae | Dol1 | 0 | 2 | 0 | 0 | 1 |
| | Empididae | Em1 | 1 | 5 | 0 | 0 | 19 |
| | Muscidae | *Limnophora* | 1 | 2 | 1 | 0 | 8 |
| | Simuliidae | *Gigantodax* | 3 | 0 | 4 | 0 | 2 |
| | | *Simulium* | 5 | 77 | 82 | 686 | 1 |
| | Tipulidae | *Hexatoma* | 2 | 4 | 4 | 0 | 5 |
| | | *Limonia* | 1 | 0 | 0 | 0 | 2 |
| | | *Molophilus* | 2 | 8 | 0 | 1 | 0 |
| | | *Tipula* | 21 | 25 | 5 | 13 | 16 |
| Ephemeroptera | Baetidae | *Andesiops* | 307 | 46 | 266 | 0 | 163 |
| | | *Baetodes* | 600 | 458 | 253 | 1 | 630 |
| | | *Camelobaetidius* | 0 | 2 | 71 | 0 | 1 |
| | | *Mayobaetis* | 24 | 6 | 7 | 0 | 14 |
| | | *Nanomis* | 11 | 43 | 0 | 0 | 1 |
| | | *Paracloeodes* | 0 | 5 | 0 | 0 | 0 |

analysis (*Neter, Wasserman & Kutner, 1990*). All statistical analysis was performed using R version 3.2.1 (*R Core Team, 2015*; Table S3, R-code, and input data in Data S1).

**Table 1** (*continued*)

| Order | Family | Genera | Ref1 | Ref2 | CP | Agr | Mi |
|-------|--------|--------|------|------|-----|-----|-----|
| | | *Prebaetodes* | 1 | 9 | 4 | 0 | 0 |
| | | *Varipes* | 0 | 11 | 0 | 0 | 0 |
| | Leptohyphidae | *Leptohyphes* | 0 | 34 | 18 | 0 | 1 |
| | | *Tricorythodes* | 0 | 12 | 0 | 0 | 0 |
| | Leptophlebiidae | *Farrodes* | 0 | 1 | 0 | 0 | 0 |
| | | *Thraulodes* | 0 | 20 | 0 | 0 | 0 |
| Hemiptera | Veliidae | *Paravelia* | 0 | 0 | 0 | 1 | 0 |
| | | *Rhagovelia* | 1 | 32 | 0 | 78 | 0 |
| Lepidoptera | Pyralidae | Cryl1 | 0 | 1 | 0 | 0 | 0 |
| Megaloptera | Corydalidae | *Corydalus* | 0 | 3 | 0 | 0 | 0 |
| Odonata | Calopterygidae | Calo1 | 0 | 3 | 0 | 22 | 0 |
| | Libellulidae | Libe1 | 0 | 18 | 1 | 3 | 0 |
| Plecoptera | Perlidae | *Anacroneuria* | 2 | 1 | 13 | 0 | 0 |
| Trichoptera | Calamoceratidae | *Phylloicus* | 1 | 0 | 0 | 0 | 0 |
| | Glossosomatidae | *Culoptila* | 5 | 5 | 27 | 0 | 1 |
| | | *Mortoniella* | 0 | 0 | 2 | 0 | 0 |
| | Helicopsychidae | *Helicopsyche* | 1 | 153 | 0 | 0 | 0 |
| | Hydrobiosidae | *Atopsyche* | 139 | 50 | 55 | 0 | 87 |
| | Hydropsychidae | *Leptonema* | 0 | 6 | 0 | 0 | 0 |
| | | *Smicridea* | 10 | 398 | 62 | 23 | 5 |
| | Hydroptilidae | *Hydroptila* | 0 | 0 | 1 | 0 | 2 |
| | | *Metrichia* | 0 | 2 | 0 | 0 | 0 |
| | Leptoceridae | *Atanatolica* | 0 | 0 | 0 | 0 | 1 |
| | | *Nectopsyche* | 4 | 1 | 4 | 0 | 1 |
| | | *Oecetis* | 0 | 0 | 0 | 1 | 0 |
| | | *Triplectides* | 0 | 1 | 0 | 0 | 0 |
| | Odontoceridae | *Marilia* | 0 | 6 | 0 | 0 | 0 |
| | Philopotamidae | *Chimarra* | 0 | 2 | 1 | 19 | 0 |
| | Polycentropodidae | *Polyplectropus* | 1 | 0 | 0 | 0 | 0 |
| Tricladida | Planariidae | *Dugesia* | 1 | 5 | 24 | 178 | 1 |
| Tubificada | Naididae | Nai1 | 14 | 0 | 3 | 8 | 6 |
| Total abundance | | | 1,506 | 2,233 | 1,366 | 1,052 | 1,326 |

**Notes.**

Ref1, Reference 1; Ref2, Reference 2; CP, Cattle production; Agr, Agriculture; Mi, Mining.

# RESULTS

A total of 7,483 organisms were collected, belonging to 14 orders, 42 families and 71 recognizable taxonomic units (RTUs), of which 57 were at genus and 14 at family level (Table 1). The stream with the greatest AMI density was Reference 2 with 1808.7 ind/m$^2$, followed by Reference 1 with 1219.8 ind/m$^2$. These were followed by the Cattle production-dominated stream with 1106.5 ind/m$^2$, then the Mining stream with 1,074 ind/m$^2$ and Agriculture stream with 852.1 ind/m$^2$. However, density was significantly higher only in the zone Reference 2 (Fr = 3.10, $df = 29$, $p$-value = 0.0163; Nemenyi *post hoc* test, $p$-value =

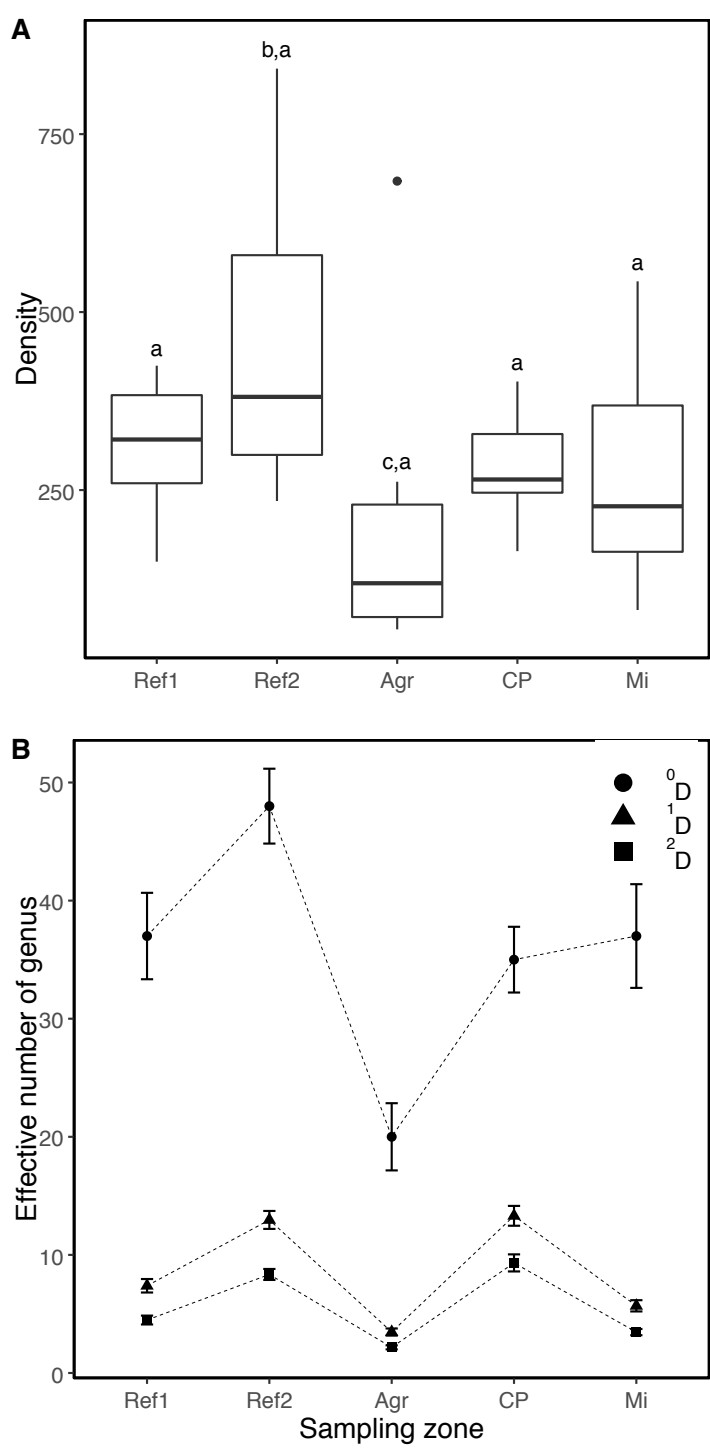

**Figure 2 Comparison of the density and diversity of aquatic macroinvertebrates (AMI) in five sampling zones.** (A) Boxplot showing the median AMI density. (B) Patterns of diversity expressions, richness ($^0$D), typical diversity ($^1$D), and effective number of the most abundant morpho-species ($^2$D). The vertical line indicates the CI 95% per $^q$D. No share letters above boxplot indicate the statistical difference between pairs of the sampling zones. Streams: Ref1, Reference 1; Ref2, Reference 2; CP, Cattle production; Agr, Agriculture; and Mi, Mining.

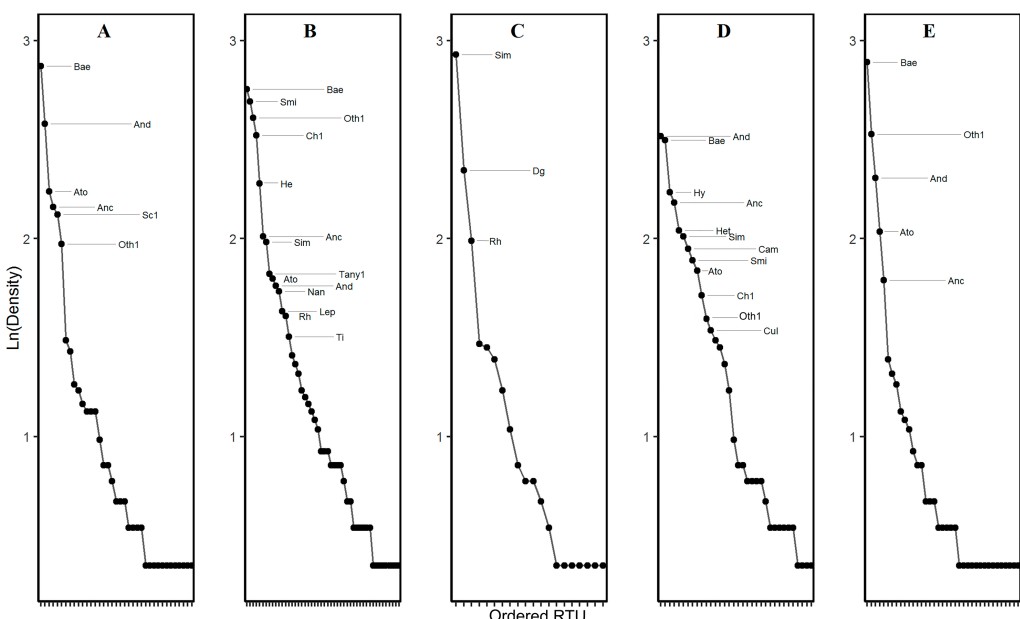

**Figure 3** **Rank–Density curve the RTUs of aquatic macroinvertebrates recorded in five sampling zone, Chichiná basin, Colombian Andes.** (A) Reference 1. (B) Reference 2. (C) Agriculture. (D) Cattle production. (E) Mining. Bae, *Baetodes*; And, *Andesiops*; Ato, *Atopsyche*; Anc, *Anchytarsus*; Sc1, , Scirtidae; Oth1, Orthocladiinae; Smi, *Smicridea*; Ch1, Chironominae: He, *Helicopsyche*; Dg, *Dugesia*; Rh, *Rhagovelia*; Hy, *Hyalella*; Cam, *Camelobaetidius*; Cul, *Culoptila*. Showed the RTUs with density larger > 25 inds * m-1.

0.0163) (Fig. 2A). In all of the sampling areas, the rank-density curves showed low equality among the communities, where less than 50% of the RTUs presented densities higher than 25 Inds/m$^2$ (i.e., dominant RTUs) (Fig. 3). Apart from the agricultural impact zone (Agr), *Baetodes* and *Anchytarsus* were common dominant RTUs among the sampling zones, in which *Baetodes* always occupied the first two positions, even in the Cattle production (CP) and Mining (Mi, Tks) streams (Fig. 3). In the Agriculture-dominated stream, only three RTUs made up the group of dominant taxa: *Simulium*, *Dugesia*, and *Rhagovelia* (Fig. 3).

According to the 95% CI, the agricultural zone presented the lowest significant values for the three expressions of diversity ($^qD$) (Fig. 2B). In contrast, the other sampling zones differed according to diversity expression. In the case of the observed richness of the RTUs ($^0D$), the zones were ordered as follows: Reference 2>Reference 1>Cattle production ≈ (Mining) (Fig. 2B). In particular, Reference 2 presented an increase in RTU richness that was between 1.3 (Ref2 vs. Ref1) and 4.3 (Ref2 vs Agr) times greater than the other sampling zones. Regarding the effective number of equally common RTUs ($^1D$), the following pattern was obtained: (Reference 2 ≈ Cattle production)>Reference 1>Mining. In this case, Reference 2 and Cattle production were between 1.3 and 3.8 times more diverse than other zones. In relation to the effective number of the most abundant RTUs, the zones were ordered in a decreasing pattern ($^2D$): Cattle production>Reference 2>Reference 1>Mining (Fig. 2B), where the magnitude of the difference ranged from 1.1 (CP vs Ref2) to 4.3 (CP vs. Agr)-fold.

No tendency of significant variation was detected in AMI density with respect to water flow (p-value: $0.18 - 0.94$) and precipitation (p-value: $0.17 - 0.82$) (Figs. S1 and S2). The physicochemical parameters of the water in the studied streams were within the quality thresholds admissible for human and domestic use (articles 38 and 39 of the Colombian Decree 1594 of 1984). The only exceptions were presented during the third sampling (July 2014), which produced values of total coliforms and *E. Coli* that exceeded admissible levels in the Agriculture stream (410,600 CFU/100 mL and 2,417 CFU/100 mL, respectively), and exceeded admissible levels for total coliforms in the Mining stream (22,470 CFU/100 ml).

Eight RTUs were shared by the five sampling zones: *Baetodes*, *Simulium*, *Anchytarsus*, *Smicridea*, *Tipula*, *Culoptila* and the subfamilies Chironominae and Orthocladiinae. The nMDS analysis evidenced separation among the different sampling streams (Fig. 4; Stress $= 0.13$), which is consistent with that found in the ANOSIM. Both tests showed that there were differences among all of the streams in terms of composition (ANOSIM: $R = 0.673$, p-value $= 0.001$). The SIMPER analysis indicated that *Baetodes*, *Simulium* and *Smicridea* were the taxa that contributed most to the differences found among the studied streams. The CCA presented an appreciable association between environmental parameters, sites and macroinvertebrates (Fig. 5: CCA1 + CCA2 = 63.2% of explained variance), where the Agricultural zone had physicochemical profiles and biotic components that were differentiated and remained separated. The Agricultural zone also presented the highest values of TS (Fig. 5) and lowest values of DO (Table S1), associated with the highest values of density of the taxa *Simulium*, *Chimarra*, *Dugesia*, *Rhagovelia* and Calopterygidae, while some Ephemeroptera and Coleoptera (*Anchytarsus* and *Heterelmis*) were practically absent from this stream (Table 1). The Cattle production and both Reference streams were associated with high values of DO, in addition to the high density of the RTUs *Baetodes*, *Mayobaetis*, *Andesiops* and *Anchytarsus* (Fig. 5; Table S1). The Mining stream, however, was strongly associated with the highest phosphate values and high values of TS, as in the Agriculture stream (Fig. 4), and presented a decrease in the majority of the previously mentioned RTUs.

## DISCUSSION

The Agricultural zone had a greater effect on AMI diversity (lowest values of richness and density) than the Mining zone, which did not follow the expected pattern in our study. These results are probably associated with the traditional horticultural practices (e.g., soil preparation and use of agrochemicals) over several years in zones of the Chinchiná river basin (Caldas, Colombia: *Meza-S et al., 2012*; *Chará-Serna et al., 2015*; *Llano, Bartlett & Guevara, 2016*); a land use situation that traditionally occurs throughout the Andes (*Mesa, 2010*; *Guevara, 2014*; *Vimos-Lojano, Martínez-Capel & Hampel, 2017*). The expansion of agricultural land use strongly reduces the presence of totally pristine headwater ecosystems in many mountainous countries (*Vimos-Lojano, Martínez-Capel & Hampel, 2017*), where several cultivated areas converge toward mainstream channels (*Chará et al., 2007*; *Chará-Serna et al., 2015*). With respect to the density, and contrary to expectation, the dominance of some RTUs tolerant to water contamination did not imply a linear increase in the total density of RTUs from the reference areas to the streams with anthropic impact.

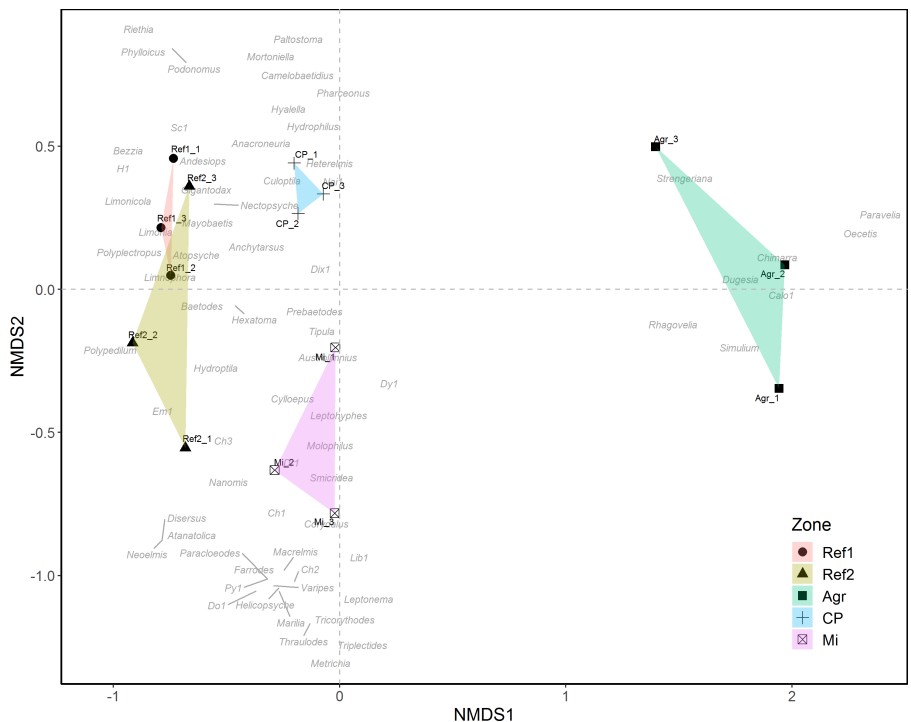

**Figure 4** **Non-Metric Multidimensional Scaling (NMDS) analysis based on the Bray–Curtis Index considering each sampling event per zone (Stress = 0.13).** The names of AMI RTUs are shown (see Table S2). Streams: Ref 1, Reference 1; Ref 2, Reference 2; CP, Cattle production; Agr, Agriculture; and Mi, Mining.

The higher AMI values of richness and density recorded in the reference and cattle production zones could be linked to the presence of riparian vegetation and its importance in buffering environmental impacts (e.g., *Lenat, 1984*; *Rivera, 2004*; *Burrdet & Watts, 2009*; *Egler et al., 2012*). However, the stream Reference 2 presented the highest values, which is possibly due to the greater differential contribution of leaf litter from speciose riparian vegetation, producing a greater availability of coarse organic benthic resources in this zone (*Gutiérrez-López, Meza-Salazar & Guevara, 2016*). It is important to note that the agricultural zone did not have riparian vegetation, which may be the reason for the lowest richness and density values found there, as is the case in other studies (e.g., *Lenat, 1984*; *Lenat & Crawford, 1994*; *Hepp et al., 2010*; *Egler et al., 2012*). Although this study was not aimed at testing the role of the riparian vegetation, this result partially coincides with the notion that removal of this vegetation can have both direct and indirect effects on AMI abundance (*Lenat, 1984*; *Egler et al., 2012*), due to the consequent degradation of both habitat and water quality (*Chará et al., 2007*). Indeed, low values of richness in zones of agriculture with similar circumstances have been previously reported by other authors (e.g., *Lenat, 1984*; *Lenat & Crawford, 1994*; *Hepp et al., 2010*; *Egler et al., 2012*), who argue that deterioration in water quality influences the number of aquatic invertebrate taxa.

The diversities $^1$D and $^2$D presented a similar pattern, due to the high importance or dominance of the most abundant RTUs in each of the studied streams. The high diversity in

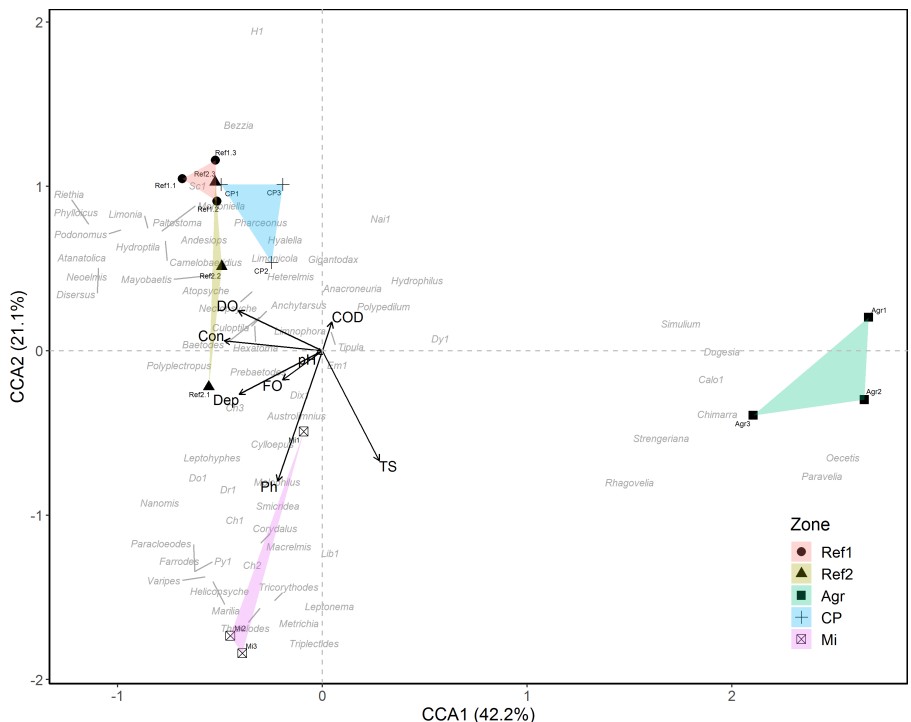

**Figure 5** **Correspondence Canonical correspondence analysis (CCA) among AMI RTUs composi-
tion and eighth hydrological, physicochemical, bacteriological (HPCB) parameters regarding measure
events per sampling zone.** The selected HPCB parameters present a VIF < 10. Streams: Ref1, Reference 1;
Ref2, Reference 2; CP, Cattle production; Agr, Agriculture; and Mi, Mining. Sampling events: 1 = Feb14
+ Apr14; 2 = Jul14 + Sept14; 3 = Nov14 + Nov15.

the Reference 2 and Cattle production streams, as well as the significantly greater diversity
in the Reference 1 compared to the Mining and Agriculture streams, could also be related
to the presence of riparian vegetation since, although the Cattle production zone does
present effects related to this activity, the strips (ca. 3 m in width) of vegetation that exist
on both sides of the stream may act to diminish these effects on the AMI community.
*Niemi & Niemi (1991)* indicate that vegetation has a positive effect on streams immersed
in cattle production zones, since it acts as a barrier to the animals and traps sediments that
are transported towards the water bodies by surface runoff. Consequently, the Mining and
Agriculture streams presented the lowest values of diversity, being significantly lower in
the Agriculture stream. These land use changes, in which riparian vegetation is replaced by
human activities such as mining and agriculture, lead to a constant alteration of the physical
characteristics of the water bodies and can thus directly or indirectly influence changes
in the spatial and/or temporal diversity of the AMI (*Tomanova & Usseglio-Polatera, 2007*;
*Domínguez & Fernández, 2009*).

For all three diversity expressions (i.e., $^0D$, $^1D$, $^2D$), the lowest values were presented
in the stream influenced by agricultural activities. *Chará-Serna et al. (2015)* reported that
one of the most important indirect consequences of agricultural practices for the AMI
community is an increase in the values of ammoniacal nitrogen ($NH_3$-N). The present

study did not find values of this parameter as high as those reported by other authors in Neotropical streams (*Mesa, 2010*; *Vázquez, Aké-Castillo & Favila, 2011*; *Chará-Serna et al., 2015*). However, *Gücker, Boëchat & Giani (2009)* explain that, although the values in streams with agriculture may be low, they still exceed those in zones with no impact. This coincides with our results, in which the values of $NH_3$-N in the Agriculture stream (0.323 mg/L) exceeded those of both Reference zones (Reference 1: 0.153 mg/L; Reference 2: 0.175 mg/L).

In the evaluated streams, the high representativity and contribution of *Baetodes*, *Andesiops*, *Simulium* and *Smicridea*, as well as the subfamily Orthocladiinae, coincide with the results of *González-G et al. (2012)* and *Meza-S et al. (2012)* in the Chinchiná river basin, in which these taxa presented a high abundance. *Baetodes*, *Simulium*, *Smicridea* and the subfamily Orthocladiinae have a wide distribution in Neotropical basins, covering broad elevational ranges (*Sganga & Angrisano, 2005*; *Sganga & Fontanarrosa, 2006*). On the other hand, the structure of the AMI communities suggests that the anthropic disturbance of the evaluated streams, except for in the Agricultural zone, has not yet crossed a point of no return. This is because of the lack of association between the dominant RTUs and a drastic reduction in the richness of RTUs, or with a phenomenon of hyperabundance of dominant RTUs (Fig. 3). This result suggests that the areas with impact from Cattle production and Mining have not yet been homogenized until limiting the availability of different resource types. However, these results should be treated with some caution, since the changes in the structure of the community of AMIs and the incidence of tolerant RTUs may reflect the effect of factors or biases in operation, rather than the specific anthropic impact. Moreover, unlike rivers in low-lying areas, Andean streams are very complex due to the topography and orography of the landscapes. The low evenness in the communities may therefore reflect the complex dynamics of mountain rivers, which include high fluctuations in flows and sediment deposition (organic and inorganic), given the high runoff rate (*Aguirre-Pabón, Rodríguez-Barrios & Ospina-Torres, 2012*; *González-G et al., 2012*).

The nMDS analysis showed a clear separation between Agriculture and the other sampled zones. This is due to the high dominance of *Simulium*, which presents lower values than other streams, as well as the absence of pollution intolerant taxa, such as *Anacroneuria*, *Marilia* and *Camelobaetidius* (*Zúñiga & Cardona, 2009*). This result demonstrates that the presence of heavy agricultural activity in the sampling zones has a strong effect on the AMI community. *Roldán & Ramírez (2008)* indicate that a river that has suffered alterations to its natural conditions through contamination processes will reflect these effects in changes to the composition and structure of its aquatic biota. Likewise, *García & Rosas (2010)* explain that agricultural activities can cause the loss of sensitive taxa, as indeed was the case in our study. The similarity between the Reference 1 and Mining streams is due to the fact that both conditions were found on the same stream (i.e., La Elvira stream). Spatial proximity between sampling sites can potentially mask the specific effect of a disturbance on the AMI community; an effect that is maximized if the sites are located on the same watercourse (*Tolonen et al., 2017*). The density and diversity of AMI in the Mining sampling point may therefore be influenced by proximity to the Reference 1

sampling site. Although our sampling design did not adequately detect the effect of spatial autocorrelation between sampling stations, the results indicated that spatial proximity does not dampen the impact of Mining on the AMI community and on the water conditions in terms of the HPCB parameters. The compositional dissimilarity between the Mining and Reference 1 sampling sites is produced by the presence of the genera reported in Reference 1, which are relatively less abundant in the Mining stream (e.g., *Smicridea*, *Andesiops* and *Nanomis*; Fig. 5). Consequently, the CCA evidenced a clear separation between Mining and Reference 1, where the former presents groups tolerant to conditions of high-water contamination by mining activity (e.g., some Chironomidae, Tipulidae and Empididae) (see *Pond et al., 2014*). These results coincide with the idea that point scales, variation in abundance or incidence of macroinvertebrate groups can be strongly modulated by the presence and availability of microhabitats (e.g., *Park, 2016*; *Burgazzi, Guareschi & Laini, 2018*).

The isolation of the Agriculture zone in the CCA, and its high values of TS (310.7 ± 209.8) and lowest values of DO (2.3 ± 0.8), reflect the negative impact of this activity on the stream and associated biota. High concentrations of TS were found in both the Agricultural and Mining streams, reducing the entry of light to the ecosystem and affecting the energy flow of the system, which lowers its productivity levels as a consequence (*Vázquez, Aké-Castillo & Favila, 2011*). Furthermore, the increase in TS is related to the sedimentation rate (*Vásquez Zapata, 2009*) and the increase in fine sediment can, in turn, be a more significant stressor to macroinvertebrate assemblages than increased nutrient concentrations, in streams around agricultural areas (*Ladrera et al., 2019*). Moreover, this variable can affect a different group of AMI, for example, taxa adapted to swim, scrape or shred, species that respire by plastron, gills and also Coleopterans dependent on a bubble or plastron to breath (*Hauer & Resh, 1996*; *Rabeni, Doisy & Zweig, 2005*; *Ladrera et al., 2019*). In contrast, invertebrates living in the mud, burrowers and filter-collectors can be favored because they feed on fine sediment.

Low DO promotes the loss of richness, increasing the density of tolerant organisms, as mentioned by *Jacobsen & Marín (2008)*. Both variables (TS and DO) could explain the high abundance of relatively tolerant filter-collector organisms such as *Simulium* and *Chimarra*, even though *Simulium* is generally associated with watercourses with a high concentration of oxygen (*Roldán, 1996*; *Domínguez & Fernández, 2009*; *Zúñiga & Cardona, 2009*; *Villada-Bedoya et al., 2017*). However, some *Simulium* species may be more tolerant than others, so it is important to advance the taxonomic knowledge of the group for identification to species level. On the other hand, predators such as Calopterygidae, *Dugesia* and *Rhagovelia* can benefit in these environments because of resource availability, as is the case with *Rhagovelia* that move over the water surface layer, breathing atmospheric oxygen and feeding on dead or dying insects. At the same time, the Calopterygidae are generally associated with substrates at the bottom of streams, where they can tolerate low concentrations of dissolved oxygen in water (*Domínguez & Fernández, 2009*).

The Cattle production and Reference 2 zones had associated high values of DO (9.3 ± 3.3 and 5.4 ± 0.63, respectively), suggesting that these were the most conserved zones in the study, with the greatest richness of species sensitive to contamination. *Zúñiga & Cardona*

*(2009)* classified *Anchytarsus* as sensitive to pollution, which is supported by our finding that this genus presented higher density in the Reference zones. Regarding Ephemeroptera, several authors indicate that the many genera in the group are sensitive to contamination (e.g., *Zedková et al., 2014*; *Akamagwuna et al., 2019*). *Buss & Salles (2007)* highlighted the importance of including the species level for the establishment of sensitivity in water quality monitoring programs. The highest phosphate ($1.2 \pm 0.62$) and TS ($394.7 \pm 210$) values found in the Mining zones indicate the deterioration that this activity can generate in aquatic ecosystems (*Wright & Ryan, 2016*), affecting the survival of some genera of macroinvertebrates (*Ramírez et al., 2018*).

In general, low values of precipitation and water flow volume were associated with high AMI densities in the studied streams. Concomitant results have been found in other small Colombian streams (*Rodríguez-Barrios et al., 2007*; *Longo et al., 2010*; *Tamaris-Turizo, Rodríguez-Barrios & Ospina-Torres, 2013*). However, we have no evidence of high variation in density related to either of these environmental variables. *Minshall & Robinson (1998)* explain that a constant climate pattern, or one of little variation, in the riparian environment translates into lower variability in the AMI dispersion dynamic. Moreover, *Smith & Lamp (2008)* suggest that the abundance and composition of the AMI community are influenced more by land use than by the seasons of high and low rains. This suggestion is consistent with the results of our study.

Despite our attempt to continuously evaluate both physicochemical and biological parameters, mining and agriculture activities present highly variable management practices (e.g., frequency and quantity of chemicals used). It is difficult to control this anthropogenic factor, which occurs jointly with natural hydrological patterns (see *Friberg, 2014*) in the selected small streams. Although these are key elements (i.e., the contribution of natural and anthropogenically-induced changes) for consideration in the patterns of stream macroinvertebrate distribution (e.g., *Domisch et al., 2017*; *Kakouei et al., 2018*), this aspect was beyond the scope of the present study due to logistical restrictions. Further studies are therefore necessary to adequately evaluate the variability of AMI due to both anthropogenic and natural pressures. It is recommended that future studies employ a larger number of spatial replicates incorporating the effects of each of the impacts and that a rigorous search of the zones of reference is conducted in order to ensure the absence of anthropogenic effects. In addition, evaluation of the heavy metals present in the sediment is recommended, since this is where their concentration is likely to be highest (e.g., *Dickson et al., 2019*).

## CONCLUSIONS

Contrary to our central hypothesis, the results show that the Agricultural zone had the lowest macroinvertebrate density and diversity. In this sense, beyond the environmental diagnosis based on physicochemical and bacteriological variables, the use of diversity measures ($^qD$) can be a useful tool to evaluate the impact of human activity on freshwater in-stream biota, since they allow adequate quantification of changes in the structure of AMI communities, using units with biological sense.

## ACKNOWLEDGEMENTS

The authors are grateful to the Universidad de Caldas, Aguas de Manizales S.A. and to members of the Bionat research group for their collaboration in the fieldwork. We also thank the anonymous reviewers and Editor for all the support and improvements provided during the reviewing process of our manuscript.

### Funding

This study was funded by the Vicerrectoría de Investigaciones y Postgrados of the Universidad de Caldas and the "Departamento Administrativo de Ciencia, Tecnología e Innovación" (Colciencias) (Project 1127-569-34668). The funders had no role in study design, data collection and analysis, decision to publish, or preparation of the manuscript.

### Grant Disclosures

The following grant information was disclosed by the authors:
Vicerrectoría de Investigaciones y Postgrados of the Universidad de Caldas.
Departamento Administrativo de Ciencia, Tecnología e Innovación: 1127-569-34668.

### Competing Interests

The authors declare there are no competing interests.

### Author Contributions

- Ana M. Meza-Salazar conceived and designed the experiments, performed the experiments, analyzed the data, prepared figures and/or tables, authored or reviewed drafts of the paper, and approved the final draft.
- Giovany Guevara and Lucimar Gomes-Dias conceived and designed the experiments, performed the experiments, authored or reviewed drafts of the paper, and approved the final draft.
- Carlos A. Cultid-Medina conceived and designed the experiments, analyzed the data, prepared figures and/or tables, authored or reviewed drafts of the paper, and approved the final draft.

### Field Study Permissions

The following information was supplied relating to field study approvals (i.e., approving body and any reference numbers):

Specimen collection permits were regulated by Resolution 1166 of October 9th, 2014, issued by the National Environmental Licenses Authority (ANLA) of Colombia and by decree 1376 of June 27th, 2013.

### Data Availability

Data and R-code are available in GitHub: https://github.com/carloscultid84/DiversityAMIs_CodeData.git.

## Supplemental Information

Supplemental information for this article can be found online at http://dx.doi.org/10.7717/peerj.9619#supplemental-information.

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
