# Peer review of "Density and diversity of macroinvertebrates in Colombian Andean streams impacted by mining, agriculture and cattle production"

_PeerJ, doi:10.7717/peerj.9619_

## Round 0.1 · original submission · Major Revisions

Both reviewers ask you to consider multivariate analyses to relate community composition with the environmental parameters (like RDA). Reviewer 2 rightly requests you assess the impact of spatial autocorrelation in your results (you could use a Moran's I tests, for example), and also suggests you include the presence of riparian vegetation as an additional covariate on your analyses. Finally, please provide your R scripts either in a GitHub (or any other open-access) repository or as supplementary material.

Reviewer 1 ·

Basic reporting

In this manuscript, the authors assessed the effects of different land uses (mining, cattle production and agriculture) on aquatic macroinvertebrate assemblages in the Colombian Andean region. The text is well written and reading is easy and consistent. I appreciate the manuscript and think the authors have good data in hands that were not well explored in the manuscript. I have some comments below:
Some references throughout the text are, in my opinion, not adequate to the referred citation. For example, in line 59 (Blanchette & Pearson 2012) don’t discuss anything about ecosystem services provided by streams. In other parts of the text, authors use local references to quote general patterns in ecology or general patterns. Also, there are too many references that could be reduced in 30-40%.

In my opinion, the weakness of the manuscript is that a novelty is not presented. What is the novelty comparing with the works cited in lines 94-100? A reduction in diversity and richness is a common and known pattern. So, the authors should go forward and show what their paper has of innovation and new knowledge to science. The authors argue that the effects of mining, agriculture and cattle production in the Andes streams within the same sampling period, but each site was analyzed separately. I suggest the authors use multivariate analyses to relate the macroinvertebrates composition/richness with the environmental parameters (CCA, RDA, Co-inertia, etc.). What parameters vary with different land uses and how it affects aquatic assemblages? It would enhance the quality of the manuscript.

Experimental design

Please see comments at "General comments".

Validity of the findings

Comments below

Additional comments

Specific comments:

L68: I suggest remove Branco 1984.

L68-70: Before going straight to mining, agricultural and cattle production as main threats to aquatic quality, I suggest describe first the main threats and impacts of aquatic ecosystems. I suggest quoting Dudgeon et al 2006 and Reid et al 2019 (Biological Reviews). Remove Sadeguian etal 2000.

L73-74: I suggest removing Zapata & Mejía 2004 and Brévault et al 2007 and add a more recent reference.

L167: Why those substrates were choice? You discuss nothing in the text (results and discussion).

L168: I suggest clarify the number of samples/subsamples: 3 replicates x 3 substrates x 6 sampling events = 54 samples/site? Is this correct?

L169: If I have well understood, authors have just one morphoespecies by genera, except for Tipulidae. I suggest using and analyzing just with genera data, once does not make sense using morphoespecies if no variability exists. Furthermore, using morphoespecies is not much adequate, once a same species can present different stages of development and be mistakenly considered different species.

L177: How water parameters were measured in situ? What equipment have you used? Also, how the aquatic parameters were measured in laboratory?

L235: In table S1 there are not 86 morphoespecies, but 57. Please check this information.

L 236-239: I suggest rewriting this sentence. “Density was significantly higher only in the zone Reference 2…”. Also, put letter (a, b) in above the boxplots showing who is different from whom.

L259: If you analyzing using morphoespecies, why for composition have you used genera?

L262: The taxon names in the figure are too small and it is difficult to read.

L263-264: Please add the statistical value of Anosim.

L269-285: And what about differences in aquatic parameters among the zones? It would be interesting testing and describing those parameters that were different among the zones. How the environmental parameters affect aquatic assemblages? You could go ahead and perform multivariate analyses, such as CCA, RDA, Coinertia, etc, to relate environmental variables and aquatic composition/richness/density.

Discussion: Authors discuss most of the results using possible explications (i.e. “could be related”) of parameters that were in fact measured by them, but not analyzed with biotic data. I think this is the weakest point of the manuscript and could be easily solved.

Table 2 is not called in the text. It could be moved to supplementary.

Reviewer 2 ·

Basic reporting

Overall, this manuscript represents an original, self-contained study evaluating the effect that different categories of anthropogenic activities purportedly have on communities of Macro-invertebrates in Colombian Andean headwaters streams. While written in completely understandable English, I spotted some confusing passages (as for example those found in line 94 or lines 121-125). As a non-native English speaker, I suggest that the manuscript should be double-checked for language inconsistencies by someone more qualified for the task, although they seemed to me as minor issues. The literature review is sufficiently presented, but with a certain regional bias to Andean South America (suggested literature to be included are for example Carter et al., 2017; Godoy et al, 2017; Abdo et al., 2013; Buss et al., 2015).
The manuscript structure is in compliance with the journal´s suggested format, as are figures and tables. Original data are presented as tables in the main body or the supplemental material sections of the manuscript, but scripts of analyses are not, which would greatly improve the reproducibility of results found in this study. I suggested authors share them as GitHub link or supplemental R_markdown file.

Suggested references:
Abdo et al. 2013, doi:10.1111/een.12013
Buss et al., 2015, doi: 10.1007/s10661-014-4132-8
Carter et al., 2017- doi:10.1016/B978-0-12-813047-6.00016-4
Godoy et al., 2017 doi:10.1007/s13744-016-0452-4

Experimental design

Regarding the experimental design of this study, the approach to deal with diversity in the analytical section is a very interesting one (a very strong reason to share the analytical scrips designed for the study). On the other hand, I would like to point out two issues that should be noted:
- Hypothesis construction in the manuscript introduction section lists several studies conducted in South America, especially Colombia, describing knowledge accumulated on the impact of broad categories of anthropogenic activities in AMI communities, but fails to clearly depict which changes or processes might be specific to each broad category. The reader is left with a very general idea of impacts that could be expected to occur in the system studied here, and a gap leading to the presented predictions. For example, is not clear why the authors expect "a maximum impoverishment of AMI diversity in the zone with gold mining" (Line 125). I suggest this could be better explained in the introduction section before any decision about the manuscript is made.

- There is no real spatial or treatment replication in the study. AMI samples are replicated only on a temporal perspective, for the same 100m strip of stream in each treatment. (In fact, they have two "control" reference sites, but they are treated as independent on the experimental design). The sampling of environmental covariates followed the same design but, except for water flow and precipitation, all other variables were not measured concomitant with AMI sampling.
In my opinion, these problems limit the inference of the impact of anthropogenic activities in AMI communities. The whole temporal replication is collapsed for the diversity analysis, allowing for an "overall" diversity comparison between each single treatment unit (mining, cattle, agriculture etc...).
Only when approaching species composition that they really allow for a true temporal evaluation, but as there is a mismatch between AMI and covariates sampling, the evaluation is restricted to an indirect comparison of ordinations (instead of a direct, more powerful approach like canonical analysis - i.e. RDA).
Also, as there are no true replicates of the treatments, the results presented here don´t allow us to discern what are anthropogenic effects and what is variability related to landscape or spatial patterns... for example "Mining" and "Reference site 1" are both located on the same stream, inside forested areas, less than 300m between them (Figure 1), and both presented very similar diversity measures (Figure 2.B) and species composition (Figure 3), and also some similarity between sampling points in covariates PCA (Figure 4). Specifically, these results would benefit from an analysis of autocorrelation in species composition between sampling points, and I suggest the authors include it at least as supplemental material in this manuscript before any decision is made.
(suggested literature: Heino, 2013; Tolonen et al., 2017).

Suggested references:
Heino, 2013, doi: 10.1002/ece3.470
Tolonen et al., 2017, doi:10.5061/dryad.2s4g5

Validity of the findings

As discussed in the previous review section, while not inconsistent with the original question of the study, the analytical approach employed here has some setbacks related to spatial replication that limits the conclusions found by the authors. I believe that the issues raised in the previous section should be addressed before a complete decision on the validity of findings.
Also, I would like to add as a note that, interestingly, the presence of riparian vegetation is mentioned in the introduction, described in the study area section and widely mentioned in the discussion section as a possible explanation to the patterns found in this study, but it was not directly evaluated in the analyses conducted. Together with spatial variables, riparian vegetation (width?) and maybe also land use classification seems to be, at least, complementary in explaining observed patterns.

Additional comments

On figure 4, the PCA plot shows "LP" for cattle production instead of "CP". The authors should correct this to maintain uniformity between legend and figure, and also across the whole manuscript

---

## Round 0.2 · Major Revisions

I would like to apologize for the delay in my response. The current pandemic made it difficult for some reviewers to allocate time to re-review your paper. As you can see Reviewer 2 made a few more (minor) suggestions but stresses out the need for extensive English proof-reading. Please address these comments in full. In addition:

- Please use colors in Fig. 1 (PeerJ papers are published online so there is no need to use black-white figures only).
- Please place Table 1 as supplementary material and bring Table S2 to the main text as a figure or summarized table.
- You mention that Spearman correlations were performed to analyze how changes in AMI density were related to flow and precipitation, and reffer to Table S1. This table does not contain any correlation coefficients, and the correlation results seem to be absent from the main text. Please clarify.

Reviewer 2 ·

Basic reporting

I noticed the authors effort in addressing some of my previous comments, but I feel that the manuscript still has language issues and has not been through a thorough revision by a native English speaker, as suggested.
In some passages, the wording is more confusing than in the previous version (as for example in lines 92-93 or 100-102, also In the discussion).

Experimental design

The authors sufficiently addressed the majority of my previous comments, except for the spatial autocorrelation issue which, as they recognize, maybe a major weakness of this study. Nonetheless, results indirectly provide evidence against spatial dependence muddling the comprehension of results obtained.

Validity of the findings

I can perceive the authors' efforts in addressing my previous comments.
One additional remark that I make relates to the fact that the predictions made on the introduction were not confirmed, while a general hypothesis of impact due to anthropogenic activities was. These predictions can be broken down into two aspects: 1) Increase in dominance due to the increase in the abundance of tolerant species, reflecting in the overall density 2) Mining as the worst possible scenario, mainly because of channel diversion and removal of organic matter and sediments.
The authors succeed in discussing the second item, but I failed to notice where they directly approach the first one, discussing why the found an opposite result than expected.

---

## Round 0.3 · accepted · Accept

I see that you have addressed all the remaining issues, so I am happy to accept your manuscript.